# Quantitative Real-Time RT-PCR Verifying Gene Expression Profile of Cavitations Within Human Jaw Bone

**DOI:** 10.3390/biomedicines13051144

**Published:** 2025-05-08

**Authors:** Shahram Ghanaati, Eva Dohle, Fabian Schick, Johann Lechner

**Affiliations:** 1FORM, Frankfurt Orofacial Regenerative Medicine, Department for Oral, Cranio-Maxillofacial and Facial Plastic Surgery, Medical Center of the Johann Wolfgang Goethe University, 60590 Frankfurt, Germany; s.ghanaati@med.uni-frankfurt.de; 2ABIS e.V. (Academy for Biological Innovations in Surgery Formally Known as SBCB e.V.), Society for Blood Concentrate and Biomaterials e.V., 60435 Frankfurt, Germany; 3Clinic for Integrative Dentistry, 81547 Munich, Germany; drfabischick@outlook.de (F.S.); drlechner@aol.com (J.L.)

**Keywords:** fatty degenerative osteonecrosis of the jaw, inflammatory mediators, quantitative real-time RT-PCR, gene expression profile, transalveolar ultrasonography

## Abstract

**Background/Objectives:** Immune cells are integral to bone homeostasis, including the repair and remodeling of bone tissue. Chronic dysregulation within this osteoimmune network can lead to bone marrow defects of the jaw (BMDJ), particularly fatty degenerative osteonecrosis of the jaw (FDOJ). These localized pathologies are implicated in systemic immune dysfunctions. **Methods:** This study is designed to determine whether BMDJ/FDOJ samples are indicative of medullary bone pathology by evaluating FDOJ gene expression patterns using quantitative real-time PCR. **Results:** Comparative analyses between pathological and healthy samples evaluated the dysregulation of key molecular pathways. BMDJ/FDOJ samples showed significant upregulation of inflammatory mediators, including CCL5/RANTES, VEGF, IGF and KOR, and downregulation of structural proteins, such as collagen types I, II and IV, and osteogenesis-associated factors, such as SP7. **Conclusions:** The study provides new insights into the molecular mechanisms of BMDJ/FDOJ by identifying potential molecular changes suggesting a pro-inflammatory state in the affected jawbone which may contribute to systemic immune dysregulation. The findings are consistent with morphologic observations of BMDJ/FDOJ in degenerated jawbone and underscore the need for integrative approaches in dentistry and medicine while highlighting BMDJ/FDOJ as a potential target for therapeutic and preventive strategies against systemic diseases and emphasizing its clinical significance.

## 1. Introduction

Immune cells play a crucial role in maintaining the delicate balance of bone homeostasis. They regulate the formation, remodeling and repair processes of bone tissue, making them indispensable in ensuring proper skeletal function. Moreover, immune cells are pivotal in facilitating wound healing by coordinating the inflammatory response and tissue regeneration. This intricate interplay between the immune system and bone tissue has given rise to the field of “osteoimmunology”, which specifically explores the dynamic relationship between bone cells, such as osteoclasts and osteoblasts, and various immune cells, including T cells, B cells and macrophages [1]. This interdisciplinary field focuses on how immune cells not only influence the development and maintenance of bone tissue but also play critical roles in bone repair, remodeling and homeostasis. Immune cells, through cytokine release and direct interactions with bone cells, can modulate bone resorption and formation, affecting both normal bone turnover and pathological conditions [2].

In the osteoimmune network, osteoclasts, responsible for bone resorption, and osteoblasts, which facilitate bone formation, interact closely with immune cells that regulate their activity [3]. T cells and B cells can influence osteoclast activity, either promoting or inhibiting bone resorption, while macrophages are involved in both bone repair and inflammation [4]. Disruptions in these immune–bone interactions can lead to bone diseases [5]. Chronic dysregulation of this osteoimmune balance has been implicated in bone marrow defects of the jaw (BMDJ). A significant manifestation of this condition is fatty degenerative osteonecrosis of the jaw (FDOJ), characterized by areas of bone necrosis accompanied by fatty tissue degeneration [6]. These chronic degenerative changes are increasingly recognized as pathological outcomes of immune system imbalances affecting the jawbone. Notably, BMDJ and FDOJ lesions are often associated with systemic immune dysregulation [7], linking them to broader pathophysiological conditions such as autoimmune disorders, chronic inflammatory diseases and metabolic syndromes. The loss of local bone marrow cells due to chronic stimulation from injuries, such as jawbone inflammation, can lead to persistent osteoimmune dysregulation [4]. Interestingly, Ghanaati et al. recently provided a new understanding of failed socket healing, considering it as a patient-specific process [8]. The authors described morphological socket changes as a combination of socket collapse and appositional bone formation resulting in the formation of a cavitation inside the socket. Ghanaati et al. pointed out in this context that these cavitational pathologies of the BMDJ/FDOJ areas are still being discussed in scientific community today, as conventional radiographic methods in dentistry do not provide adequate imaging. As a complementary noninvasive imaging approach, transalveolar ultrasonography (TAU) has been developed to objectively detect BMDJ/FDOJ cavities without the use of ionizing radiation [9]. In previous studies, we have characterized this chronic inflammatory process in BMDJ/FDOJ and described the associated pathological morphology [10]. BMDJ/FDOJ is primarily defined in the literature as “bone marrow oedema” [11] or as a silent/subclinical inflammatory condition that lacks the typical signs of acute inflammation. Our clinical experience with the notable fatty degenerative morphology of BMDJ/FDOJ led us to investigate the cytokine expression in those BMDJ/FDOJ tissue samples easily excised from such defects. From a panel of 27 cytokines analyzed for samples obtained from five patients, we found the unique overexpression of interleukin−1 receptor antagonist (IL-1ra) and the proinflammatory chemokine CCL5/RANTES (RANTES = Regulated And Normal T cell Expressed and Secreted) [12]. In contrast, TNF-α expression, which was expected to be high, presented vanishingly low levels. This imbalance between primarily surprisingly high levels of Th-2 cytokines IL-1ra and CCL5/RANTES brought us to the hypothesis that BMDJ/FDOJ areas might be connected to the epistemology of different diseases. Interestingly, the data support the involvement of CCL5/RANTES in the pathogenesis of colorectal cancer (CRC) and indicate its potential value as a therapeutic target [13]. A potential correlation between the expression level of CCL5/RANTES and mutational burden has been demonstrated [14].

Advancements in molecular and cellular analysis techniques, particularly quantitative real-time PCR (qRT-PCR), might provide deeper insights into gene expression changes associated with jawbone defects. This methodology allows us to precisely assess and verify gene expression profiles linked to osteoimmune dysregulation, reinforcing the role of BMDJ and FDOJ in systemic immune dysfunction. Multiplex cytokine and chemokine analyses have already demonstrated that chronic inflammatory mediators in BMDJ/FDOJ lesions could contribute to broader immune imbalances [10,11,13,14].

Surgical intervention to remove BMDJ/FDOJ tissue has been proposed to mitigate systemic effects, while transalveolar ultrasonography (TAU) is emerging as a promising diagnostic tool to overcome the limitations of radiographic imaging [8,15,16,17]. By elucidating the pathways through which immune dysregulation contributes to localized bone defects, researchers might be able to develop innovative strategies for prevention and treatment. While osteoimmunology is the interdisciplinary field that explores the interaction between the immune system and the skeletal system, we focused on factors associated with the process of osteogenic differentiation and inflammation in general as well as on factors connected to immune dysregulation that might contribute to localized bone defects. The aim of the present study is to evaluate the FDOJ relative gene expression patterns of 22 osteoimmunology-related factors using quantitative real-time PCR. This growing field of research holds promise for improving clinical outcomes in patients with chronic osteoimmune imbalances, ultimately advancing the field of regenerative medicine and bone health, particularly in maxillofacial and orofacial contexts.

## 2. Materials and Methods

In order to elucidate the previously unknown and widely discussed [18] underlying molecular basis, we examined in a first step the histology of three BMDJ/FDOJ samples in comparison with one healthy jawbone sample and in a second step the gene expression profile of different inflammatory mediators and structural and osteoblastic factors of three different morphologically pathological samples of BMDJ/FDOJ and, as a control, three healthy primary osteoblasts samples using the quantitative real-time PCR method.

### 2.1. Collection of the Jawbone Samples

In total, 3 different samples of FDOJ tissue and 3 different samples of healthy tissue have been analyzed for histology. Healthy tissue samples have been received as anonymous excess tissue samples prior to dental implantation (exploratory drilling with trephine drill). Samples used for this study were collected from FDOJ patients in the Clinic for Integrative Dentistry, Munich. In this clinic, samples were assessed as pathological using transalveolar ultrasonography (TAU) as a diagnostic tool according to previous published work [16,17]. When the tissue has been diagnosed as pathological, the tissue has been removed. These excess tissue samples were sent anonymously from the Clinic for Integrative Dentistry, Munich, for histological and gene expression analyses to FORM, Frankfurt. The study was performed in accordance with the Declaration of Helsinki and the study was approved by the responsible Ethics Commission of the state Hessen, Germany (special vote 2024–2023; 9 September 2024). Samples used for this study, collected from FDOJ patients, were of a similar size (~3 mm × 3 mm) and from the same location. For histology of FDOJ tissue samples, in total, 3 different pathological samples (donors) were processed and stained. For qRT-PCR experiments, 3 different pathological samples (donors) were processed for estimation of gene expression profile.

### 2.2. Histology of FDOJ Tissue Samples

After fixation of the healthy and FDOJ tissue samples in formalin (4% paraformaldehyde; Carl Roth, Karlsruhe, Germany), the samples were transferred into embedding cassettes. Subsequently, 3 morphologically distinct pathological samples of BMDJ/FDOJ tissue were processed with a tissue processor (Leica, Wetzlar, Germany) and thus prepared for the embedding process. First, a dehydration step with a series of different alcohol concentrations (70%, 96%, 96%, 100%, 100%) was performed for 45 min to 1 h each. The tissue samples were then soaked in xylene (VWR, Darmstadt, Germany) 3 times for 1 h and embedded in paraffin. After completion of the paraffin embedding process, the samples were sectioned with a rotary microtome and placed in a heating oven for at least 12 h. For each sample, 5 slides with a thickness of 2 μm were prepared. Finally, the slides were stained using AZAN and Masson Goldner (Carl Roth, Karlsruhe, Germany) staining.

### 2.3. Isolation of Healthy Human Primary Osteoblasts (pOBs)

The pOBs were isolated from healthy bone tissue obtained as surplus material (excess tissue) during surgical procedures. The bone was manually minced into small fragments using sterile bone scissors and thoroughly rinsed with phosphate-buffered saline (PBS; Thermo Scientific, Karlsruhe, Germany) to remove blood and debris.

Bone fragments were then transferred into 6-well culture plates and incubated in Dulbecco’s Modified Eagle Medium/Nutrient Mixture F-12 (DMEM/F-12, Ham’s; Sigma-Aldrich, St. Louis, MI, USA) supplemented with 20% fetal calf serum (FCS, Sigma-Aldrich, St. Louis, MI, USA) and 1% penicillin–streptomycin (Sigma-Aldrich, St. Louis, MI, USA). Cultures were maintained at 37 °C in a humidified incubator under an atmosphere of 95% air and 5% CO_2_. Over the course of 2 to 4 weeks, osteoblast-like cells migrated out from the bone explants and proliferated across the surface of the wells. Once a sufficient outgrowth was observed, the cells were detached using 0.25% trypsin-EDTA (Sigma-Aldrich, St. Louis, MI, USA) and subsequently transferred to T-75 cell culture flasks for expansion and further processed for RNA isolation and cDNA synthesis.

### 2.4. Analysing Gene Expression Profile in BMDJ/FDOJ Tissue Samples

In order to elucidate the previously unknown network of the underlying molecular basis, we examined 3 morphologically distinct pathological samples of BMDJ/FDOJ by quantitative real-time PCR, comparing BMDJ/FDOK RNA versus RNA from healthy human primary osteoblasts (pOBs) as a control providing a deeper understanding of the molecular processes that regulate bone formation and resorption.

#### 2.4.1. Sample Preparation

RNA was isolated from the tissue pieces using an established protocol (TRIzol RNA isolation method): First, 1 mL of TRIzol reagent (Thermo Scientific, Karlsruhe, Germany) was added to each tissue piece and was incubated for 5 min to permit complete dissociation of the nucleoprotein complex before tissue samples were homogenized using a ball mill. After adding 0.2 µL chloroform per 1 mL TRIzol reagent and incubating for 5 min, samples were centrifuged for 15 min at max speed (12,000× *g*) before the aqueous phase containing RNA was transferred to a new tube followed by two washing steps using isopropanol and ethanol. Finally, after max speed centrifugation followed by discarding the supernatants, the remaining RNA pellet was air dried and resuspended in 10 µL of water. RNA concentration was measured using a nanodrop. RNA was used when the A260/A280 ratio was between 1.8 and 2.0. A total of 1 µg of extracted RNA was used to transcribe into complementary DNA (cDNA) according to a standard protocol using Omniscript Reverse Transcription Kit (Qiagen, Hilden, Germany). As such, 2 µL Reverse Transcription buffer (RT buffer), 2 µL Random Primer (50 µM stock), 2 µL nucleoside triphosphate (dNTPs, 10 mM stock), 1 µL Reverse Transcription Omniscript enzyme, 1 µL RNase inhibitor and 10 µL template (1 µg of RNA diluted in RNase-free water) were used. The mixture was incubated at 37 °C for 1 h.

#### 2.4.2. Quantitative Real-Time PCR

Quantitative real-time PCR, enabling the quantification of relative gene expression, was performed using SYBR green DNA-binding fluorescent dye. A total of 12.5 µL of QuantiTectTM SYBR^®^ Green PCR Master Mix (Qiagen, Hilden, Germany), 2.5 µL of QuantiTectTM SYBR^®^ Green primer assay (10X stock, Qiagen, Hilden, Germany), 6 µL of RNase-free water and 4 µL of cDNA (1 ng/µL) were used for one reaction. Quantitative real-time PCR was performed in triplicates with the following cycler program:

95 °C 15 min, denaturation step: 94 °C 15 s, annealing step: 55 °C 30 s, elongation step: 72 °C 35 s; dissociation: 95 °C 15 s, 60 °C 1 min, 95 °C 15 s, 40 cycles performed in total. The relative gene expressions of 22 factors were estimated during this study (Figure 1 results). Ribosomal protein 37A (RPL37A) was taken as an endogenous standard, and relative gene expression was determined using the ∆∆Ct method. Primers were commercially purchased from Qiagen (Quantitect primer assays, Qiagen, Hilden, Germany; Appendix A). Primer specificity has been evaluated by performing a melting curve analysis post PCR to confirm the amplification of a single, specific product. Gene expression was compared by setting control cultures to 1 (reference value) as indicated in the relevant tables and figures and data are additionally presented as logarithmic scale.

## 3. Results

### 3.1. Histological Evaluation of BMDJ/FDOJ Samples

Both AZAN and Masson Goldner staining can be used for analyzing BMDJ/FDOJ pathology by providing a detailed view of degenerative and necrotic changes in jawbone tissue while assessing bone integrity and collagen degradation (Figure 1). Both histological stainings provides clear differentiation between collagen, muscle and bone matrix components. Compared to a healthy control bone tissue sample (C/F), the stainings of BMDJ/FDOJ samples clearly reveal a loss of mineralized bone and the existence of fatty infiltration in affected jawbone regions (A/B/D/E).

**Figure 1 biomedicines-13-01144-f001:**
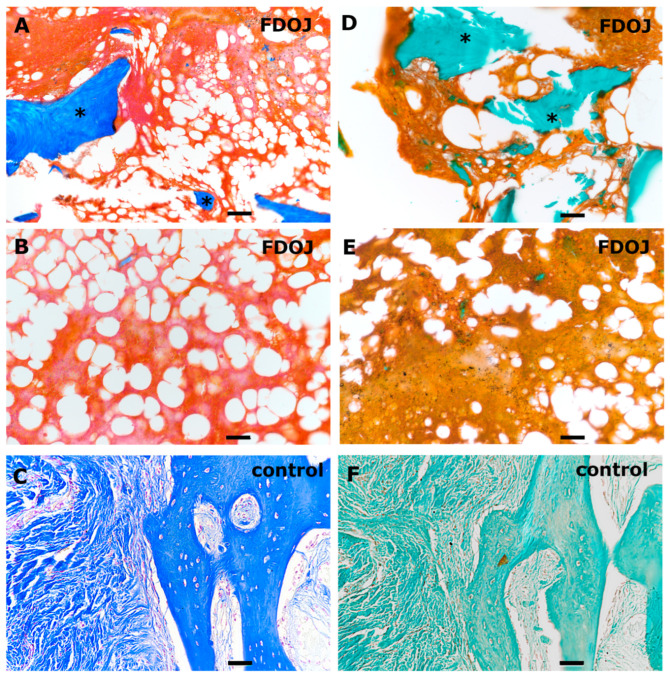
Histological evaluation of BMDJ/FDOJ tissue samples (**A**,**B**,**D**,**E**) compared to healthy control bone tissue (**C**,**F**). Samples were stained using AZAN (**A**–**C**) and Masson Goldner (**D**–**F**). Scale bar 50 µm. The asterisks in (**A**,**D**) highlight few calcified bone structures (blue/green) in FDOJ tissue samples compared to controls (**C**,**F**).

### 3.2. Relative Gene Expression Profile of BMDJ/FDOJ RNA Versus Healthy pOB RNA

The results of the quantitative real-time PCR analysis of three morphologically distinct pathological samples of BMDJ/FDOJ and, as a control, three healthy primary osteoblasts samples are presented as logarithmic scale (Figure 2). Table 1 shows the results listed as relative quantification (RQ) in tabular form.

Since the focus of our investigation is on the differences in gene expression from healthy reference values (pOB) to BMDJ/FDOJ samples, we compared the existing deviations of each factor and summarized them into two groups, namely, overexpressed (upregulated gene expression) factors in BMDJ/FDOJ and underexpressed (downregulated gene expression) factors in BMDJ/FDOJ. Comparing the gene expression profiles of 22 different genes involved in osteoimmunological processes in FDOJ tissues and healthy osteoblasts, we could evaluate the inflammatory mediators CCL5/RANTES, IGF (Insulin-like growth factor), Flt1 (Vascular endothelial growth factor receptor 1) and KOR (kappa opioid receptor) as significantly upregulated in FDOJ tissue samples. On the other hand, BMDJ/FDOJ samples showed significant downregulation of osteogenic factors and structural proteins, such as collagen types I, II and IV, and osteogenesis-associated factors, such as SP7 (transcription factor). BMDJ/FDOJ relative quantification of gene expression values from 1.4 to 0.6 were rated as balanced and were not interpreted during this study. BMP2 (Bone morphogenetic protein 2; RQ = 1.114592), ICAM (Intercellular adhesion molecule 1; RQ = 0.24263), SELP (P-Selectin; RQ = 0.441399), Col10 (Collagen 10; RQ = 0.641733), Col14 (Collagen 14; RQ = 0.397088), Col18 (Collagen 18; RQ = 1.478216), Lama (Laminin A; RQ = 0.274826), OC (Osteocalcin; RQ = 0.394182).

## 4. Discussion

Bone marrow defects of the jaw (BMDJ) and fatty degenerative osteonecrosis of the jaw (FDOJ) are chronic, asymptomatic and often overlooked pathological conditions characterized by ischemic bone degeneration and fatty infiltration in the jawbone [6]. Accordingly, histological evaluation of BMDJ/FDOJ jawbone samples could confirm pathological changes and highlight fatty tissue alterations and loss of bone tissue compared to healthy controls [6]. Emerging research suggests that BMDJ/FDOJ may have systemic implications beyond localized bone pathology, particularly in its potential to dysregulate immune function [7,10,11,13,14,19]. The presence of inflammatory mediators in necrotic jawbone areas has been linked to chronic immune activation and systemic inflammatory responses [7,10,11,13,14,19]. This connection raises concerns about the role of BMDJ/FDOJ as a hidden source of chronic inflammation that may contribute to immune dysregulation, autoimmune conditions and systemic diseases. Understanding the interplay between BMDJ/FDOJ and immune system dysfunction is crucial for identifying potential diagnostic markers and therapeutic interventions for patients with unexplained chronic inflammatory conditions. During this study, BMDJ/FDOJ samples indicative of medullary bone pathology were used to evaluate BMDJ/FDOJ gene expression patterns using quantitative real-time PCR. The comparative analyses between pathological and healthy samples evaluated the dysregulation of key molecular pathways at the gene expression level.

The comparative gene expression analyses revealed a significant upregulation of inflammatory mediators, including CCL5/RANTES, VEGF, IGF and KOR. CCL5/RANTES is expressed and secreted by normal T cells and functions as a classical chemotactic cytokine as well as acting as a proinflammatory chemokine that recruits leukocytes to sites of inflammation. It exhibits chemotactic activity for T cells, eosinophils, basophils, monocytes, natural killer (NK) cells, dendritic cells and mastocytes [20]. CCL5/RANTES is primarily expressed by T cells and monocytes [19]. Gene expression analyses of CCL5/RANTES could be determined as significantly upregulated in BMDJ/FDOJ tissue compared to healthy osteoblasts. In fatty degenerative osteonecrosis of the jaw (FDOJ), CCL5/RANTES is significantly upregulated, contributing to a persistent inflammatory state. Elevated CCL5/RANTES levels in BMDJ/FDOJ lesions can promote immune dysregulation, enhance osteoclastic activity and sustain chronic low-grade inflammation, potentially linking BMDJ/FDOJ to systemic inflammatory and autoimmune conditions [6,7,21]. This excessive inflammatory signaling may also interfere with normal bone healing and vascularization, exacerbating the degenerative changes characteristic of BMDJ/FDOJ. The strong association between CCL5/RANTES upregulation and BMDJ/FDOJ highlights its potential as both a biomarker and a therapeutic target for managing the condition and its systemic effects. CCL5/RANTES plays a key role in sustaining inflammation and induces the expression of matrix metalloproteinases, which are crucial for cell migration into inflamed tissues [22]. CCL5/RANTES has been proven to confirm a pathological marrow structure in the BMDJ/FDOJ areas. In previous multiplex studies, we could already show that T-cell-induced CCL5/RANTES overexpression is a characteristic signaling pathway for BMDJ/FDOJ bone pathologies [6,7,10,12,15]. Overexpression of CCL5/RANTES seems to be the osteoimmunological fingerprint of BMDJ/FDOJ. Earlier publications on a possible immunological systemic effect of CCL5/RANTES [11,13,14] are supported by scientific research: CCL5/RANTES is involved in transplantations [23] and the development of tumors [24]. Furthermore, CCL5/RANTES is involved in numerous human diseases and disorders, such as viral hepatitis or COVID-19 [25]. In a recent publication, we developed epidemiologically relevant effects of chronic CCL5/RANTES vehiculation in BMDJ/FDOJ on acute COVID-19 CCL5/RANTES cytokine storms [25]. FLT1 (Fms-related tyrosine kinase 1), also known as VEGFR-1, is a receptor involved in vascular signaling and inflammation [26], [27]. Upregulation of FLT1 in BMDJ/FDOJ tissue samples suggests a link between impaired vascularization, chronic inflammation and immune dysregulation. In BMDJ/FDOJ lesions, ischemic conditions and oxidative stress may trigger increased FLT1 expression, which can contribute to dysfunctional angiogenesis and promote a pro-inflammatory microenvironment. This persistent inflammation may enhance the secretion of inflammatory cytokines, such as CCL5/RANTES, further influencing systemic immune responses [28,29,30]. The overexpression of FLT1 in BMDJ/FDOJ could therefore also serve as a biomarker for chronic inflammatory bone conditions and help explain the broader systemic effects associated with jawbone ischemia and necrosis. Insulin-like growth factor 1 (IGF-1) plays a critical role in normal growth and development by modulating cell proliferation, differentiation, glucose and lipid metabolism and cell survival. Insulin-like growth factor 1 (IGF1) also plays a crucial role in bone metabolism, tissue repair and immune regulation [31,32]. In BMDJ/FDOJ, IGF1 upregulation may indicate a compensatory response to chronic ischemia and bone degeneration [33]. However, despite increased IGF1 expression, the necrotic and fatty degenerated bone in BMDJ/FDOJ fails to regenerate properly, suggesting impaired signaling pathways or a dysfunctional bone-healing environment. Additionally, IGF1 can interact with inflammatory cytokines, potentially influencing immune dysregulation in BMDJ/FDOJ lesions [34,35]. This upregulation may contribute to the persistence of a low-grade inflammatory state, linking BMDJ/FDOJ to systemic chronic inflammation and immune-related disorders [36,37,38]. The liver is the primary source of circulating IGF-1, producing it in response to growth hormone [39]. IGF-1 is the predominant growth factor in the bone matrix, playing a crucial role in bone homeostasis across all life stages [31,32]. It regulates longitudinal bone growth, skeletal maturation and bone mass acquisition during preadolescence while also being essential for maintaining bone mass in adults [33]. The kappa opioid receptor (KOR) is involved in pain modulation, immune regulation and inflammatory responses [40]. In BMDJ/FDOJ, altered KOR expression may play a role in the disease’s asymptomatic nature and its impact on the immune system [41]. KOR activation can suppress pain signaling, potentially explaining why BMDJ/FDOJ often remains undetected despite ongoing bone degeneration and chronic inflammation [7,8,16,17]. Additionally, KOR influences immune function by modulating cytokine release, which may contribute to the dysregulated inflammatory environment observed in BMDJ/FDOJ lesions. This connection suggests that KOR activity in BMDJ/FDOJ could play a dual role—masking pain while promoting an inflammatory microenvironment that may have systemic immune consequences. The strong expression of the kappa opioid receptor (KOR) in the BMDJ/FDOJ areas is striking. The former term for this condition is ‘neuralgia-inducing osteonecrosis of the jaw/NICO’ [12] by BMDJ/FDOJ. Their involvement in numerous cases of atypical facial pain and trigeminal neuralgia [42,43] corresponds to this KOR protein expression. This is a key factor in the modulation of trigeminal pain perception. The relationship between BMDJ/FDOJ and the pain-inducing blockade of opioid receptors has already been pointed out in earlier publications [44,45].

On the other hand, the downregulation of osteogenic factors (such as Receptor Activator of NF-κB Ligand (RANKL), BMPs and OPG) and structural proteins (like collagen and osteocalcin) contributes to impaired bone regeneration and progressive degeneration in BMDJ/FDOJ [46,47]. This reduction in key bone-forming molecules leads to weakened bone matrix integrity, decreased mineralization and fatty infiltration of the jawbone. The compromised structural environment fosters chronic inflammation and ischemia, creating a self-perpetuating cycle of bone degradation [48]. Additionally, the loss of osteogenic signaling may contribute to immune dysregulation by altering the balance of pro-inflammatory and anti-inflammatory factors in the affected bone tissue, further linking BMDJ/FDOJ to systemic inflammatory and immune-related conditions. RANKL was found to be downregulated in terms of gene expression level in BMDJ/FDOJ tissue. RANKL is a protein from the tumor necrosis factor (TNF) family and is significantly involved in the regulation of bone remodeling [46,47]. The RANK/RANKL system is a biochemical regulatory circuit that ensures that bone resorption remains in a healthy balance with bone formation, which is a prerequisite for the dynamic architecture of the bone system [47]. In BMDJ/FDOJ, an imbalance in RANKL expression can contribute to excessive bone resorption and impaired healing. The chronic under-expression of RANKL, which is essential for harmonious bone metabolism, proves that the balanced regulation of bone remodeling in the BMDJ/FDOJ areas is defective. Additionally, RANKL is involved in immune regulation and inflammation [47], suggesting that its dysregulation in BMDJ/FDOJ may contribute to chronic inflammatory signaling and systemic immune disturbances. This connection highlights RANKL’s potential role in the progression of BMDJ/FDOJ and its broader impact on bone and immune system health [47]. Alpha Smooth muscle Actin (SMA) is a family of globular multi-functional proteins that form microfilaments in the cytoskeleton, the thin filaments in muscle fibrils, and is a key marker of myofibroblasts, which are involved in wound healing, fibrosis and vascular regulation [48]. Actin networks give mechanical support to cells and provide trafficking routes through the cytoplasm to aid signal transduction [49]. In BMDJ/FDOJ tissue, the downregulated SMA expression may indicate impaired blood flow and chronic ischemia within the affected bone. In BMDJ/FDOJ, dysfunctional or reduced myofibroblast activity may contribute to chronic ischemia, impaired angiogenesis and defective bone healing. Since myofibroblasts play a crucial role in maintaining microcirculation [49], their dysfunction in BMDJ/FDOJ could lead to inadequate oxygen and nutrient delivery, promoting bone degeneration and fatty infiltration. Additionally, their impaired function in BMDJ/FDOJ may contribute to the persistent inflammatory environment and immune dysregulation seen in affected jawbone areas. This connection underscores the role of vascular health in BMDJ/FDOJ progression and its broader systemic effects. A downregulation of SMA causes a diminished signal transduction in the alveolar tissue and restricted cytoplasmic metabolic pathways [49]. This is consistent with the chronically blocked self-healing tendency and autoregulation leading to successful wound healing in BMDJ/FDOJ areas. Bone morphogenetic proteins (BMPs) are multi-functional growth factors. BMP1 induces bone and cartilage development. BMP1 is an essential enzyme involved in extracellular matrix formation and collagen maturation, playing a critical role in bone development and repair [50]. The here-evaluated BMP1 downregulation in BMDJ/FDOJ tissue may contribute to defective bone remodeling, impaired collagen synthesis and weakened structural integrity of the jawbone. This disruption can lead to increased bone fragility, fatty infiltration and chronic ischemia, all of which characterize BMDJ/FDOJ lesions [6]. Additionally, BMP1 influences the balance between bone formation and resorption, meaning its dysregulation may further contribute to the inflammatory and osteolytic environment seen in BMDJ/FDOJ. This connection highlights BMP1’s role in the impaired healing processes and chronic degenerative changes associated with BMDJ/FDOJ. SP7 (Sp7 Transcription Factor) is a protein-coding gene that regulates bone development [51,52] and was also shown to be downregulated in terms of gene expression level in BMDJ/FDOJ tissue. Diseases associated with SP7 include Osteogenesis Imperfecta [53]. It plays a major role, along with Runx2 and Dlx5, in driving the differentiation of mesenchymal precursor cells into osteoblasts and osteocytes [51]. In adult mice, ablation of Sp7 led to a lack of new bone formation, highly irregular cartilage accumulation beneath the growth plate and defects in osteocyte maturation and functionality [54]. Along similar mechanistic lines as bone repair is the integration of dental implants into alveolar bone, since the insertion of these implants causes bone damage that must be healed before the implant is successfully integrated [55]. Researchers have shown that when bone marrow stromal cells are exposed to artificially elevated levels of Sp7/Osx, mice with dental implants were shown to have better outcomes through the promotion of healthy bone regeneration [55]. New bone formation is reduced in SP7 deficiency, which corresponds to the stasis of osteoneogenesis in BMDJ/FDOJ. Furthermore, if implants are placed in a BMDJ/FDOJ area with under-expressed SP7, the protein dysfunction demonstrated in our study will prevent successful osseointegration of the implant [55]. We have previously demonstrated an associated local osteoimmunological risk for the long-term systemic defense status due to only moderately osseointegrated implants [56]. Collagen-related proteins such as Col1, Col2 and Col4 are vital for strengthening and promoting the formation of all connective tissue parts of the body, including the bones [57]. The morphology of BMDJ/FDOJ is a clear indication of reduced connective tissue formation and a possible consequence of chronic underactivity of collagen activities as Col1, Col2 and Col4 were found to be downregulated in BMDJ/FDOJ tissue samples at gene expression level. Collagen-related proteins such as Col1, Col2 and Col4 are vital for strengthening and promoting the formation of all connective tissue parts of the body, including the bones [58]. Furthermore, the reduced gene expression of osteonectin (ON) in BMDJ/FDOJ tissues also means reduced calcium binding and thus reduced supply to the osteoblasts [59]. Osteonectin (ON) is a glycoprotein in the bone that binds calcium. It is secreted by osteoblasts during bone formation, initiating mineralization and promoting mineral crystal formation, and plays a vital role in bone mineralization, cell-matrix interactions and collagen binding [60]. This protein is synthesized by macrophages at sites of wound repair and platelet degranulation, so it may play an important role in wound healing [61]. ON also increases the production and activity of matrix metalloproteinases, a function important to invading cancer cells within bone [61]. This limits the important role of ON in wound healing and in the osseointegration of dental implants in the jaw. As a result, the BMDJ/FDOJ medullary bone defect lacks adequate mineralization [7,11,62].

Although this study provides valuable insights into the molecular mechanisms of BMDJ/FDOJ, there are several limitations to consider. The sample size and number of analyzed donors might limit the generalizability of the findings to a larger population and might also potentially overlook confounding factors such as age, gender and comorbidities, which could influence the gene expression results. Furthermore, while the study focused on key inflammatory and structural markers, it did not explore other relevant molecular pathways, such as immune cell interactions or oxidative stress markers, which could have provided a more comprehensive view of the disease. Additionally, the use of quantitative PCR to measure gene expression does not capture post-transcriptional modifications or protein-level changes, limiting the interpretation of the findings and making future research necessary to address these limitations and enhance the understanding of BMDJ/FDOJ and its broader implications for systemic diseases. Nevertheless, this study is original in its investigation of the possible molecular mechanisms behind BMDJ/FDOJ and their potential connection to systemic immune dysfunctions. By analyzing gene expression patterns in pathological jawbone samples, it has uncovered significant changes in inflammatory mediators and structural proteins, shedding new light on how immune dysregulation can lead to bone marrow defects. This study also highlights the potential of BMDJ/FDOJ as a target for therapeutic strategies, not only addressing local jawbone degeneration but also broader systemic diseases. Its integrative approach bridging dentistry and medicine, combined with the use of quantitative PCR to examine gene expression, offers a novel perspective on understanding bone pathology and its wider implications.

## 5. Conclusions

BMDJ/FDOJ samples show a different gene expression pattern compared to healthy controls. Comparative analyses of pathological and healthy samples revealed different gene expression profiles and dysregulation of key molecular pathways. BMDJ/FDOJ samples exhibited significant upregulation of inflammatory mediators, alongside the downregulation of structural proteins as well as osteogenesis-related factors like SP7. These findings align with the observed morphological degeneration in BMDJ/FDOJ-affected jawbones and emphasize the need for integrative approaches in dentistry and medicine. Although further research is needed, the study provides new insights into the molecular mechanisms of BMDJ/FDOJ and identifies potential therapeutic targets for further investigations. Recognizing BMDJ/FDOJ as a potential target for therapeutic and preventive strategies could play a crucial role in addressing its broader implications for systemic diseases.

## 6. Patents

The corresponding author is the holder of two patents used in the TAU apparatus CaviTAU^®^ PCT/EP2018/084199 and PCT/EP2020/058962.

## Figures and Tables

**Figure 2 biomedicines-13-01144-f002:**
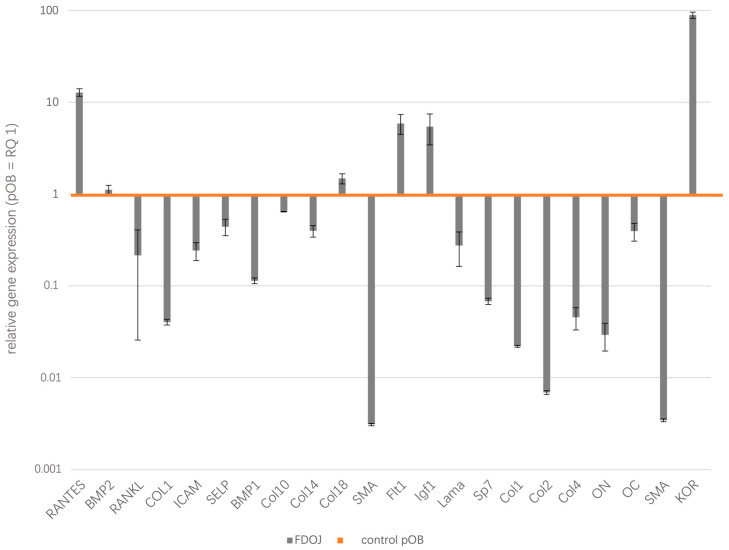
The relative gene expression (RQ) of 22 factors were estimated during this study, providing significant insights into the molecular mechanisms connected to BMDJ/FDOJ. Ribosomal protein 37A (RPL37A) was taken as an endogenous standard and relative gene expression was determined using the ∆∆Ct method. Gene expression was compared by setting control cultures to 1 (healthy pOB as reference value, orange line) as indicated and data are presented as logarithmic scale.

**Table 1 biomedicines-13-01144-t001:** The relative gene expression of 22 factors (RQ) in BMDJ/FDOJ tissue presented and listed in tabular form. Ribosomal protein 37A (RPL37A) was taken as an endogenous standard and relative gene expression was determined using the ∆∆Ct method. Gene expression was compared by setting control cultures to 1 (healthy pOB as reference value) as indicated. (FDOJ = gene expression in FDOJ tissue; control = gene expression in healthy control primary osteoblasts).

Gene	FDOJ	Control	Gene	FDOJ	Control
RANTES	12.79 ± 1.2	1	Flt1	5.89 ± 1.44	1
BMP2	1.11 ± 0.13	1	Igf1	5.43 ± 2.01	1
RANKL	0.21 ± 0.19	1	Lama	0.27 ± 0.112	1
COL1	0.04 ± 0.003	1	Sp7	0.06 ± 0.0054	1
ICAM	0.24 ± 0.054	1	Col1	0.02 ± 0.0006	1
SELP	0.44 ± 0.09	1	Col2	0.00 ± 0.00032	1
BMP1	0.11 ± 0.008	1	Col4	0.04 ± 0.0125	1
Col10	0.64 ± 0.004	1	ON	0.02 ± 0.0098	1
Col14	0.39 ± 0.057	1	OC	0.39 ± 0.087	1
Col18	1.47 ± 0.187	1	SMA	0.00 ± 0.00012	1
SMA	0.00 ± 0.00009	1	KOR	89.01 ± 7.12	1

## Data Availability

The data that support the findings of this study are available on request from the corresponding author (E.D.).

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
