# Peer review of "Quantitative Real-Time RT-PCR Verifying Gene Expression Profile of Cavitations Within Human Jaw Bone"

_biomedicines, 2025, doi:10.3390/biomedicines13051144_

Round 1
Reviewer 1 Report
Comments and Suggestions for Authors
The manuscript reports on the experiment comparing the expression of selected cytokines in jawbone samples from BMDJ/FDOJ patients vs. healthy individuals. The authors wish to elucidate the pathways through which immune dysregulation contributes to localized bone defects to facilitate the development of innovative strategies for preventing and treating the BMDJ/FDOJ disorder. The research provided some novel findings. I have the following comments and suggestions for revision consideration.
Title:
The “PCR” in the title should be RT-PCR. (The gene analysis was actually qRT-PCR. Please correct this throughout the text)
Introduction:
What were the criteria for selecting those cytokines for the study?
Line 63 indicates 27 cytokines, and line 80 says 22 factors—inconsistent?
Materials and methods:
The descriptions of sample collection and preparation are somewhat vague:
Were the samples used for histology study and RT-PCR analysis from the same patients?
Were the jawbone samples collected from the patients and healthy individuals (controls) at a similar size and location?
Please explain what poB is.
Most content in Section 2.3 should be moved to discussion.
Line 127: Please specify the approximate size of the tissue piece used for RNA isolation.
Line 130: Please specify the max speed (g-force) used.
Line 137: Please indicate the concentration of primer and dNTP used for reactions.
Please provide PCR primer information.
Fig 1: The scale bar is almost invisible. Please provide a more detailed explanation /description of what to compare between BMDJ/FDOJ and the healthy samples.
Please provide the standard deviation in Figure 2 and Table 1.
Can pathway analyses be performed to reveal the dysregulated molecular pathways based on the qRT-PCR data from this study to support the claim in the Conclusion stating, “Comparative analyses of pathological and healthy samples revealed dysregulation of key molecular pathways”?
Comments on the Quality of English Language
English is acceptable. Some parts of the materials and methods need to be revised.
Author Response
Dear Reviewer,
we would like to thank you for carefully reading the manuscript and for your response. Please find included the resubmission of the manuscript with the revised title:
“Quantitative real time RT-PCR verifying gene expression profile of cavitations within human jaw bone”
by Shahram Ghanaati, Eva Dohle, Fabian Schick, and Johann Lechner
to be considered for publication as original research paper in Biomedicine. We have revised the manuscript according to your suggestions. The changes are highlighted in yellow color in the revised version of the manuscript and addressed in this letter. We would like to thank you for all your effort with the manuscript.
Yours sincerely,
Eva Dohle
General information:
The individual answers to the reviewer’s suggestions are addressed in this letter point by point. All changes in the revised manuscript have been highlighted in yellow colour.
Reviewer 1
The manuscript reports on the experiment comparing the expression of selected cytokines in jawbone samples from BMDJ/FDOJ patients vs. healthy individuals. The authors wish to elucidate the pathways through which immune dysregulation contributes to localized bone defects to facilitate the development of innovative strategies for preventing and treating the BMDJ/FDOJ disorder. The research provided some novel findings. I have the following comments and suggestions for revision consideration.
Title:
The “PCR” in the title should be RT-PCR. (The gene analysis was actually qRT-PCR. Please correct this throughout the text)
According to this suggestion, the title has been changed. In addition, we corrected the term RT-PCR in qRT-PCR throughout the whole text.
Introduction:
What were the criteria for selecting those cytokines for the study?
The selected and analyzed factors during this study are related to osteoimmunology processes. These factors include factors that are associated with the process of osteogenic differentiation and inflammation in general as well as factors connected to immune dysregulation that might contribute to localized bone defects. According to the reviewers’ question, we added an additional sentence describing the criteria for selecting the factors.
Line 63 indicates 27 cytokines, and line 80 says 22 factors—inconsistent?
Line 63 “From a panel of 27 cytokines analyzed for samples obtained from five patients, we found the unique overexpression of interleukin-1 receptor antagonist (IL-1ra) and the proinflammatory chemokine CCL5/RANTES…..” refers to a previous publication (Lechner, J. and W. Mayer, Immune messengers in Neuralgia Inducing Cavitational Osteonecrosis (NICO) in jaw bone and systemic interference. European Journal of Integrative Medicine, 2010. 2(2): p. 71-77.). The appropriate reference has been added to the main text, respectively.
Materials and methods:
The descriptions of sample collection and preparation are somewhat vague:
The authors revised the paragraph and added more information on the preparation of the samples for the different experiments as indicated in the revised manuscript.
Were the samples used for histology study and RT-PCR analysis from the same patients?
All samples used for this study were collected from FDOJ patients and were from a similar size and from the same location. Samples were assessed as pathologically using transalveolar ultrasonography (TAU) as diagnostic tool according to previous published work [1, 2]. The tissue samples were sent anonymously from the Clinic for Integrative Dentistry, Munich, for histological and gene expression analyses to FORM, Frankfurt. Three of them were processed for the gene expression study, and three were prepared and processed for histology. The authors added more precise information on that in the m&m section.
Please explain what poB is.
Accordingly, “pOB” is now explained in the m&m section of the revised manuscript.
Most content in Section 2.3 should be moved to discussion.
The sentences have been removed from the m&m paragraph, accordingly.
Line 127: Please specify the approximate size of the tissue piece used for RNA isolation.
The approximate size of the tissue pieces has been added.
Line 130: Please specify the max speed (g-force) used.
The maximum speed (g-force) has been added, accordingly.
Line 137: Please indicate the concentration of primer and dNTP used for reactions.
The primers that were used for this study are commercially purchased QuantiTect Primer Assays (Qiagen) for SYBR Green real-time PCR, which are company-tested pre-validated primer sets provided at a final stock concentration of 10x per primer (forward and reverse). For real-time PCR reactions, the primers were used at a final concentration of 1x per primer per reaction mix as Qiagen recommends. The authors added the missing information in the revised manuscript.
Please provide PCR primer information.
The authors provided a primer list including catalog numbers as supplementary table (table S1) according to this suggestion.
Fig 1: The scale bar is almost invisible. Please provide a more detailed explanation /description of what to compare between BMDJ/FDOJ and the healthy samples.
According to this suggestion, the scale bars in figure 1 have been increased in size for better visualization. Furthermore, asterisks have been added to figure 1 A/D to exemplarily show the very few calcified bone structure (blue/green) in the FDOJ tissue samples compared to the healthy control tissue (calcified bone=blue/green).
Please provide the standard deviation in Figure 2 and Table 1.
Standard deviation has been added to figure 2 and table 1, accordingly.
Can pathway analyses be performed to reveal the dysregulated molecular pathways based on the qRT-PCR data from this study to support the claim in the Conclusion stating, “Comparative analyses of pathological and healthy samples revealed dysregulation of key molecular pathways”?
Pathway analyses can be performed based on qRT-PCR data to support the claim that key molecular pathways and factors are dysregulated in pathological samples compared to healthy ones. This type of analysis might help to provide deeper insights into the molecular mechanisms underlying the appropriate disease. Since the authors have a huge experience with the pathological changes in FDOJ patients in general, the idea of this study was to identify possible underlying mechanisms in a first step. Initial studies on the potential molecular mechanisms of various pathologies might serve as a crucial foundation for further research. They allow for the preliminary identification of relevant molecular pathways, gene expression patterns, or biomarkers that may be associated with a particular disease. Although these early investigations do not include more detailed analyses or extensive experimental validation of individual findings, they still provide valuable hypotheses and key insights for future studies and might help to identify promising targets or mechanisms that can later be examined in more comprehensive and focused investigations. Thus, such initial studies are an essential step in biomedical research, as they might pave the way for future investigations, generate new scientific questions, and ultimately contribute to a deeper insight and understanding of disease mechanisms. According to this question and suggestion, we revised the conclusion part and tried to express the conclusion a bit more carefully.
References
- Lechner, J., B. Zimmermann, and M. Schmidt, Focal Bone-Marrow Defects in the Jawbone Determined by Ultrasonography-Validation of New Trans-Alveolar Ultrasound Technique for Measuring Jawbone Density in 210 Participants. Ultrasound Med Biol, 2021. 47(11): p. 3135-3146.
- Lechner, J., et al., Ultrasound Sonography to Detect Focal Osteoporotic Jawbone Marrow Defects Clinical Comparative Study with Corresponding Hounsfield Units and RANTES/CCL5 Expression. Clin Cosmet Investig Dent, 2020. 12: p. 205-216.

Reviewer 2 Report
Comments and Suggestions for Authors
The paper Quantitative real-time PCR verifying the gene expression profile of jawbone defects debates whether BMDJ/FDOJ samples indicate medullary bone pathology by evaluating FDOJ gene expression patterns using quantitative real-time PCR.
The abstract should be structured into headings.
Keywords: try not to use abbreviations
The aim of the paper should be at the end of the introduction, lines 81-89 should be moved above.
Figure 1: the caption should better explain the images: FDOJ and control, and lines and arrows are suggested as well. The color or the images should be explained and its importance enhanced.
Table 1 should have an explaining row at the start as well as a bottom explanation of the abbreviations
Limitations should be provided.
The originality of the study must be highlighted.
What are the clinical relevance and clinical implications?
The patents should be presented as a supplementary material.
Author Response
Dear Reviewer,
we would like to thank you for carefully reading the manuscript and for your response. Please find included the resubmission of the manuscript with the revised title:
“Quantitative real time RT-PCR verifying gene expression profile of cavitations within human jaw bone”
by Shahram Ghanaati, Eva Dohle, Fabian Schick and Johann Lechner
to be considered for publication as original research paper in Biomedicine. We have revised the manuscript according to your suggestions. The changes are highlighted in yellow color in the revised version of the manuscript and addressed in this letter. We would like to thank you for all your effort with the manuscript.
Yours sincerely,
Eva Dohle
General information:
The individual answers to the reviewer’s suggestions are addressed in this letter point by point. All changes in the revised manuscript have been highlighted in yellow colour.
Reviewer 2
The paper Quantitative real-time PCR verifying the gene expression profile of jawbone defects debates whether BMDJ/FDOJ samples indicate medullary bone pathology by evaluating FDOJ gene expression patterns using quantitative real-time PCR.
The abstract should be structured into headings.
According to this suggestion, the authors structured the abstract into headings.
Keywords: try not to use abbreviations
The abbrevations have been removed from he keywords.
The aim of the paper should be at the end of the introduction, lines 81-89 should be moved above.
Accordingly, the authors revised the end of the introduction.
Figure 1: the caption should better explain the images: FDOJ and control, and lines and arrows are suggested as well. The color or the images should be explained and its importance enhanced.
According to this suggestion, asterisks have been added to figure 1 A/D to exemplarily show the very few calcified bone structure (blue/green) in the FDOJ tissue samples compared to the healthy control tissue (calcified bone = blue/green). In addition, the scale bars in figure 1 have been increased in size for better visualization and the colour of the images have been explained in the figure legend.
Table 1 should have an explaining row at the start as well as a bottom explanation of the abbreviations
The table has been revised. Abbrevations of the experimental groups have been added in the table heading and abbrevations of the specific genes can be found in the abbreviation list.
Limitations should be provided.
We have added a paragraph concerning limitations of the study, accordingly.
The originality of the study must be highlighted.
According to this suggestion, the authors revised the manuscript while trying to highlight the originality in a better way.
What are the clinical relevance and clinical implications?
The clinical relevance and implications of this study are significant for both diagnosing and treating BMDJ/FDOJ and are a first step towards understanding its potential connections to systemic immune dysfunctions. The identification of specific gene expression changes in inflammatory mediators and structural proteins might provide valuable molecular biomarkers for early diagnosis of BMDJ/FDOJ, improving detection and treatment outcomes in the future.
The patents should be presented as a supplementary material.
According to the author guidelines of the Journal, patents should be reported in paragraph 6.

Reviewer 3 Report
Comments and Suggestions for Authors
The paper titled "Quantitative real-time PCR verifying gene expression profile of jawbone defects" investigates the gene expression patterns associated with Bone Marrow Defects of the Jaw (BMDJ) and Fatty-Degenerative Osteonecrosis of the Jaw (FDOJ). The study uses quantitative real-time PCR (qRT-PCR) to compare gene expression in pathological jawbone samples with healthy controls.
Here are my comments:
Page 2, line 92. The study uses a small number of samples(only 3 samples), which may not be representative of the broader population affected by BMDJ/FDOJ. Larger sample sizes could provide more robust findings.
Page 3, line 98. The data were collected as part of the everyday medical care of the patients and evaluated retrospectively. The clinical case studies we present here were realized as part of a case-control study and had a retrospective character. The study does not mention the demographic or clinical diversity of the samples. Furthermore, the criteria for patient inclusion/exclusion are not well defined.
Page 4, line 155. The paper does not provide detailed statistical analysis, such as p-values or confidence intervals, for the gene expression data. Therefore, how can we assess the significance of the findings? Maybe because of small sample size? The comprehensive statistical analysis, including p-values, confidence intervals, and effect sizes, will strengthen the validity of the results.
Page 6, line 219. This connection raises concerns about the role of BMDJ/FDOJ as a hidden source of chronic inflammation that may contribute to immune dysregulation, autoimmune conditions, and systemic diseases. The authors extrapolate the findings without sufficient evidence. Any studies to support this claim? This study only provides preliminary evidence. However, the authors could suggest that further research is needed to explore the systemic implications of BMDJ/FDOJ, particularly through longitudinal studies or animal models.
Page 10, line 392. The discussion does not adequately address the limitations of the study, such as the small sample size or potential biases in sample selection. This exclusion makes it difficult to assess the reliability and generalizability of the findings. The authors should include a dedicated section discussing the limitations of the study and how these limitations might affect the interpretation of the results. For example, they could acknowledge that the small sample size limits the generalizability of the findings, and that functional validation is needed to confirm the biological relevance of the observed gene expression changes.
Page 10, line 395. The Conclusion section of the paper summarizes the key findings but has some weaknesses. The authors suggest that BMDJ/FDOJ could be a "potential target for therapeutic and preventive strategies against systemic diseases," but the study does not provide evidence to support this claim. The authors should also highlight the novelty and significance of their findings in the conclusion. For example, they could emphasize that the study provides new insights into the molecular mechanisms of BMDJ/FDOJ and identifies potential therapeutic targets for further investigation
Comments on the Quality of English LanguageThe English could be improved to more clearly express the research.
Author Response
Dear Reviewer,
we would like to thank you for carefully reading the manuscript and for your response. Please find included the resubmission of the manuscript with the revised title:
“Quantitative real time RT-PCR verifying gene expression profile of cavitations within human jaw bone”
by Shahram Ghanaati, Eva Dohle, Fabian Schick and Johann Lechner
to be considered for publication as original research paper in Biomedicine. We have revised the manuscript according to your suggestions. The changes are highlighted in yellow color in the revised version of the manuscript and addressed in this letter. We would like to thank you for all your effort with the manuscript.
Yours sincerely,
Eva Dohle
Reviewer 3
The paper titled "Quantitative real-time PCR verifying gene expression profile of jawbone defects" investigates the gene expression patterns associated with Bone Marrow Defects of the Jaw (BMDJ) and Fatty-Degenerative Osteonecrosis of the Jaw (FDOJ). The study uses quantitative real-time PCR (qRT-PCR) to compare gene expression in pathological jawbone samples with healthy controls.
Here are my comments:
Page 2, line 92. The study uses a small number of samples (only 3 samples), which may not be representative of the broader population affected by BMDJ/FDOJ. Larger sample sizes could provide more robust findings.
While the sample size may limit generalizability of the study, the findings might offer a starting point for understanding the key inflammatory and structural changes involved in this disease. These early observations might guide future, larger-scale studies and might help to identify critical biomarkers for diagnosis, treatment, and prevention. The study's innovative approach also might pave the way for more targeted therapeutic strategies and an integrative approach to patient care, making it an important contribution to the field, even with the limitations in sample size. According to this suggestion, we added a paragraph concerning the limitations of the study in the discussion part.
Page 3, line 98. The data were collected as part of the everyday medical care of the patients and evaluated retrospectively. The clinical case studies we present here were realized as part of a case-control study and had a retrospective character. The study does not mention the demographic or clinical diversity of the samples. Furthermore, the criteria for patient inclusion/exclusion are not well defined.
Yes, the samples/data were collected as part of the everyday medical care of the patients. All samples used for this study were collected from FDOJ patients from a similar size and from the same location. Samples were assessed as pathologically using transalveolar ultrasonography (TAU) as diagnostic tool according to previous published work [1, 2]. The tissue samples were sent and analysed anonymously for histological evaluation and for gene expression studies. The authors added more precise information on that in the m&m section.
Page 4, line 155. The paper does not provide detailed statistical analysis, such as p-values or confidence intervals, for the gene expression data. Therefore, how can we assess the significance of the findings? Maybe because of small sample size? The comprehensive statistical analysis, including p-values, confidence intervals, and effect sizes, will strengthen the validity of the results.
Given the sample size of 3 donors, we chose not to perform statistical analysis in this study. Our primary goal was to highlight the observed trends in gene expression profiles within FDOJ tissue samples compared to healthy controls rather than to draw definitive conclusions about statistical significance. The focus was on identifying potential molecular markers and patterns that may warrant further investigation in larger, more statistically powered studies. While we acknowledge the limitations of the sample size, this exploratory approach serves as a valuable starting point for future research that could confirm and expand upon these initial findings. According to the reviewer’s suggestion, we revised figure 2 and table 1 and added the standard deviation of the analysed donors.
Page 6, line 219. This connection raises concerns about the role of BMDJ/FDOJ as a hidden source of chronic inflammation that may contribute to immune dysregulation, autoimmune conditions, and systemic diseases. The authors extrapolate the findings without sufficient evidence. Any studies to support this claim? This study only provides preliminary evidence. However, the authors could suggest that further research is needed to explore the systemic implications of BMDJ/FDOJ, particularly through longitudinal studies or animal models.
Thank you for this suggestion. You are correct in noting that the study provides only preliminary evidence regarding the potential role of BMDJ/FDOJ as a source of chronic inflammation contributing to immune dysregulation, autoimmune conditions, and systemic diseases. The authors’ extrapolation of these findings to broader systemic implications may be premature without sufficient supporting evidence. Nevertheless, research on other localized chronic inflammatory conditions, such as periodontitis or osteoarthritis, suggests that persistent local inflammation can influence systemic immune responses and contribute to broader health issues. Given the early-stage nature of the study, the authors revised the manuscript and suggest that further research is necessary to establish a clearer connection between BMDJ/FDOJ and systemic health.
Page 10, line 392. The discussion does not adequately address the limitations of the study, such as the small sample size or potential biases in sample selection. This exclusion makes it difficult to assess the reliability and generalizability of the findings. The authors should include a dedicated section discussing the limitations of the study and how these limitations might affect the interpretation of the results. For example, they could acknowledge that the small sample size limits the generalizability of the findings, and that functional validation is needed to confirm the biological relevance of the observed gene expression changes.
We have added a paragraph concerning limitations of the study, accordingly.
Page 10, line 395. The Conclusion section of the paper summarizes the key findings but has some weaknesses. The authors suggest that BMDJ/FDOJ could be a "potential target for therapeutic and preventive strategies against systemic diseases," but the study does not provide evidence to support this claim. The authors should also highlight the novelty and significance of their findings in the conclusion. For example, they could emphasize that the study provides new insights into the molecular mechanisms of BMDJ/FDOJ and identifies potential therapeutic targets for further investigation
According to this suggestion, the authors revised the manuscript while trying to highlight the originality, novelty and significance in the discussion part as well as in the conclusion of the study.
References
- Lechner, J., B. Zimmermann, and M. Schmidt, Focal Bone-Marrow Defects in the Jawbone Determined by Ultrasonography-Validation of New Trans-Alveolar Ultrasound Technique for Measuring Jawbone Density in 210 Participants. Ultrasound Med Biol, 2021. 47(11): p. 3135-3146.
- Lechner, J., et al., Ultrasound Sonography to Detect Focal Osteoporotic Jawbone Marrow Defects Clinical Comparative Study with Corresponding Hounsfield Units and RANTES/CCL5 Expression. Clin Cosmet Investig Dent, 2020. 12: p. 205-216.

Reviewer 4 Report
Comments and Suggestions for Authors
Dear Authors,
The manuscript presents an interesting and relevant study on the gene expression profile of bone marrow defects of the jaw (BMDJ) and fatty-degenerative osteonecrosis of the jaw (FDOJ). The use of quantitative real-time PCR (qRT-PCR) provides valuable insights into molecular dysregulation in affected tissues. However, some aspects of the manuscript require revision for clarity, consistency, and adherence to scientific rigor.
The abstract is well-structured but should emphasize the clinical implications of the findings more clearly. Consider clarifying how the gene expression changes contribute to systemic immune dysregulation.
The introduction effectively outlines the background but should define the key terms (e.g., BMDJ, FDOJ) earlier for clarity. The reference to "osteoimmune network" could be expanded with a brief explanation of the underlying immunological mechanisms. The study's hypothesis should be explicitly stated at the end of the introduction.
The methodology is well-detailed but lacks clarity in some areas: Specify the sample size more explicitly. More details on RNA quality control measures and normalization strategies are needed. The qRT-PCR protocol should include a statement on primer specificity validation.
In the results: The histological findings should be better linked to the gene expression results. Figures and tables need clearer labeling and legends. Statistical analysis of the gene expression differences should be presented more explicitly (e.g., p-values, confidence intervals).
Discussion: The discussion appropriately highlights the role of inflammatory mediators but could be more structured. I suggest to compare findings with existing literature in a more systematic manner. Also it would be better to address potential limitations of the study, such as sample size and possible confounding factors.
The conclusion should reinforce the key findings and their translational relevance to clinical practice.
Comments on the Quality of English LanguageSome sentences are overly complex and should be restructured for readability. Avoid redundancy, particularly in the discussion where certain points are repeated.
Author Response
Dear Reviewer,
we would like to thank you for carefully reading the manuscript and for your response. Please find included the resubmission of the manuscript with the revised title:
“Quantitative real time RT-PCR verifying gene expression profile of cavitations within human jaw bone”
by Shahram Ghanaati, Eva Dohle, Fabian Schick and Johann Lechner
to be considered for publication as original research paper in Biomedicine. We have revised the manuscript according to your suggestions. The changes are highlighted in yellow color in the revised version of the manuscript and addressed in this letter. We would like to thank you for all your effort with the manuscript.
Yours sincerely,
Eva Dohle
Reviewer 4
Dear Authors,
The manuscript presents an interesting and relevant study on the gene expression profile of bone marrow defects of the jaw (BMDJ) and fatty-degenerative osteonecrosis of the jaw (FDOJ). The use of quantitative real-time PCR (qRT-PCR) provides valuable insights into molecular dysregulation in affected tissues. However, some aspects of the manuscript require revision for clarity, consistency, and adherence to scientific rigor.
The abstract is well-structured but should emphasize the clinical implications of the findings more clearly. Consider clarifying how the gene expression changes contribute to systemic immune dysregulation.
According to this suggestion, the abstract has been revised.
The introduction effectively outlines the background but should define the key terms (e.g., BMDJ, FDOJ) earlier for clarity. The reference to "osteoimmune network" could be expanded with a brief explanation of the underlying immunological mechanisms. The study's hypothesis should be explicitly stated at the end of the introduction.
We revised the manuscript according to this suggestion.
The methodology is well-detailed but lacks clarity in some areas: Specify the sample size more explicitly. More details on RNA quality control measures and normalization strategies are needed. The qRT-PCR protocol should include a statement on primer specificity validation.
According to this suggestion, more detailed information on RNA quality control as well as on primer specifity validation have been added in the revised version of the manuscript. The primers that were used for this study are commercially purchased QuantiTect Primer Assays (Qiagen) for SYBR Green real-time PCR, which are company-tested pre-validated primer sets. Primer specifity has been evaluated by performing a melting curve analysis post-PCR to confirm the amplification of a single, specific product. The authors added the missing information in the revised manuscript. In addition, the authors provided a primer list as supplementary table (table S1).
In the results: The histological findings should be better linked to the gene expression results. Figures and tables need clearer labeling and legends. Statistical analysis of the gene expression differences should be presented more explicitly (e.g., p-values, confidence intervals).
According to this suggestion, we revised the manuscript.
Asterisks have been added to figure 1 A/D to exemplarily show the very few calcified bone structure (blue/green) in the FDOJ tissue samples compared to the healthy control tissue (blue/green). In addition, the scale bars in figure 1 have been increased in size for better visualization and the colour of the images have been explained in the figure legend.
Given the sample size of 3 donors, we chose not to perform statistical analysis in this study. Our primary goal was to highlight the observed trends in gene expression profiles within FDOJ tissue samples compared to healthy controls rather than to draw definitive conclusions about statistical significance. The focus was on identifying potential molecular markers and patterns that may warrant further investigation in larger, more statistically powered studies. While we acknowledge the limitations of the sample size, this exploratory approach serves as a valuable starting point for future research that could confirm and expand upon these initial findings. According to the reviewer’s suggestion, we revised figure 2 and table 1 and added the standard deviation of the analysed donors.
Discussion: The discussion appropriately highlights the role of inflammatory mediators but could be more structured. I suggest to compare findings with existing literature in a more systematic manner. Also it would be better to address potential limitations of the study, such as sample size and possible confounding factors.
We revised the discussion part and added a paragraph concerning limitations of the study, accordingly.
The conclusion should reinforce the key findings and their translational relevance to clinical practice.
According to this suggestion, the authors revised the manuscript while trying to highlight the originality, novelty and significance in the discussion part as well as in the conclusion of the study.

Round 2
Reviewer 3 Report
Comments and Suggestions for Authors
The authors have satisfactorily addressed all comments
Comments on the Quality of English LanguageThe English could be improved to more clearly express the research.
Author Response
Dear Reviewer,
we would like to thank you for carefully reading the manuscript and for your response. Please find included the resubmission of the manuscript with the revised title:
“Quantitative real time RT-PCR verifying gene expression profile of cavitations within human jaw bone”
by Shahram Ghanaati, Eva Dohle, Fabian Schick, and Johann Lechner
to be considered for publication as original research paper in Biomedicines. We have revised the manuscript again. Changes are highlighted in yellow color in the revised version of the manuscript. We would like to thank you for all your effort with the manuscript.
Yours sincerely,
Eva Dohle
Reviewer 4 Report
Comments and Suggestions for Authors
All requested changes were made and authors replied to all my questions and suggestions.
Author Response
Dear Reviewer,
we would like to thank you for carefully reading the manuscript and for your response. Please find included the resubmission of the manuscript with the revised title:
“Quantitative real time RT-PCR verifying gene expression profile of cavitations within human jaw bone”
by Shahram Ghanaati, Eva Dohle, Fabian Schick, and Johann Lechner
to be considered for publication as original research paper in Biomedicines. We have revised the manuscript according to the comments of the academic editor. The changes are highlighted in yellow color in the revised version of the manuscript and addressed in this letter. We would like to thank you for all your effort with the manuscript.
Yours sincerely,
Eva Dohle